# Building Footprint Extraction from High-Resolution Images via Spatial Residual Inception Convolutional Neural Network

**Penghua Liu** [1,2], **Xiaoping Liu** [1,2], **Mengxi Liu** [1,2], **Qian Shi** [1,2,*], **Jinxing Yang** [3], **Xiaocong Xu** [1,2] and **Yuanying Zhang** [1,2]

1   School of Geography and Planning, Sun Yat-Sen University, West Xingang Road, Guangzhou 510275, China; liuph3@mail2.sysu.edu.cn (P.L.); liuxp3@mail.sysu.edu.cn (X.L.); liumx23@mail2.sysu.edu.cn (M.L.); xuxiaocong@mail.sysu.edu.cn (X.X.); zhangyy257@mail2.sysu.edu.cn (Y.Z.)
2   Guangdong Key Laboratory for Urbanization and Geo-simulation, Sun Yat-Sen University, West Xingang Road, Guangzhou 510275, China
3   School of Geographical Sciences, Guangzhou University, West Waihuan Street/Road, Guangzhou 510006, China; yangjx11@gzhu.edu.cn
*   Correspondence: shixi5@mail.sysu.edu.cn

**Abstract:** The rapid development in deep learning and computer vision has introduced new opportunities and paradigms for building extraction from remote sensing images. In this paper, we propose a novel fully convolutional network (FCN), in which a spatial residual inception (SRI) module is proposed to capture and aggregate multi-scale contexts for semantic understanding by successively fusing multi-level features. The proposed SRI-Net is capable of accurately detecting large buildings that might be easily omitted while retaining global morphological characteristics and local details. On the other hand, to improve computational efficiency, depthwise separable convolutions and convolution factorization are introduced to significantly decrease the number of model parameters. The proposed model is evaluated on the Inria Aerial Image Labeling Dataset and the Wuhan University (WHU) Aerial Building Dataset. The experimental results show that the proposed methods exhibit significant improvements compared with several state-of-the-art FCNs, including SegNet, U-Net, RefineNet, and DeepLab v3+. The proposed model shows promising potential for building detection from remote sensing images on a large scale.

**Keywords:** semantic segmentation; high-resolution image; building footprints extraction; fully convolutional network; multi-scale contexts

---

## 1. Introduction

As the carrier of human production and living activities, buildings are of vital significance to the living environment of human beings and are good indicators to characterize the population aggregation, energy consumption intensity, and regional development [1–3]. Thus, accurate building extraction from remote sensing images will be advantageous to the investigation of dynamic urban expansion and population distribution patterns [4–8], updating the geographical database, and many other aspects. With the rapid development in remote sensing for earth observation, especially the improvement in the spatial resolution of imagery, more elaborate spectra, structure, and texture information of objects are increasingly being represented on the imagery; these make the accurate detection and extraction of buildings possible.

Nevertheless, because of the spatial diversity of buildings, including shape, materials, spatial size, and interference of building shadows, the development of accurate and reliable building

extraction methods has become an enormous problem [9]. In previous decades, several studies on building extraction, such as pixel-based methods, object-based methods, edge-based methods, and shadow-based methods, were proposed. For example, Huang et al. [10] proposed the morphological building index (MBI) to automatically detect buildings from high-resolution imagery. The basic principle of the MBI is the utilization of a set of morphological operations to represent the intrinsic spectral-structural properties of buildings (e.g., brightness, contrast, and size). In order to consider the inference of shadow of the building, Ok et al. [11] modeled the spatial relationship of buildings and their shadows by means of a fuzzy landscape generation method. Huang [12] proposed the morphological shadow index to detect shadows beside buildings; such shadows could introduce a constraint in the spatial distribution of buildings. To solve the diversity in spatial size, Belgiu and Dragut [13] proposed multi-resolution segmentation methods to consider the multi-scale of buildings. Chen et al. [14] proposed edge regularity indices and shadow line indices as new features of building candidates obtained from segmentation methods to refine the boundary of detected results. However, these studies were based on handcrafted features, so prior experience is necessary to be able to design specific features. Unfortunately, the methods with high demand for prior experience could not be widely utilized in complex situations where building diversity is extremely high.

In recent years, the developments in deep learning, especially the convolutional neural network (CNN), has promoted a new paradigm shift in the field of computer vision and remote sensing image processing [15]. By learning rich contextual information and extending the receptive field, CNNs can automatically learn hierarchical semantically related representations from the input data without any prior knowledge [16]. With the continuous development of computational capability and the explosive increase in available training datasets, some typical CNN models, such as AlexNet [17], VGGNet [18], GoogLeNet [19], ResNet [20], and DenseNet [21], have achieved state-of-the-art (SOTA) results on various object detection and image classification benchmarks. Compared with traditional handcrafted features, high-level representations extracted by CNNs have been proved to exhibit superior capability in remote sensing image classifications [22], such as building extraction [23]. However, these methods mainly focus on the label assignment of images but lack accurate positioning and boundary characterization of objects in images [24]. Therefore, in order to detect objects and their spatial distribution, the prediction of pixel-wise labels is of considerable significance [21].

Since 2015, the fully convolutional network (FCN) proposed by Long et al. [25] has attracted considerable interests in the use of CNNs for dense predictions. The FCN architecture abnegates full connections and instead adopts convolutions and transposed convolutions, which are applicable to the segmentation of images of any size. In the FCN, feature maps with high-level semantics but low resolutions are generated by down-sampling features using multiple pooling or convolutions with strides [26]. To recover the location information and obtain fine-scaled segmentation results, down-sampled feature maps are up-sampled using the transposed convolutions, and thereafter skip-connected with shallow features having higher resolutions [25]. Based on the FCN paradigm, a variety of SOTA FCN variants has been gradually proposed to exploit the contextual information for more accurate and refined segmentations. To recover the object details better, the encoder-decoder architecture was generally considered to be an effective FCN schema [26]. The SegNet [27] and U-Net [24] are two typical architectures with elegant encoder-decoder structures. Typically, there are multiple shortcut connections from the encoder to the decoder to fuse multi-level semantics. The subsequent FCNs are mainly designed to improve segmentation accuracy by extending the receptive field and learning multi-scale contextual information. For example, Yu and Koltun [28] proposed the use of dilated convolutions of different dilations to aggregate multi-scale contexts. The DeepLab networks, including v1 [29], v2 [30], v3 [31], and v3+ [26], proposed by Chen et al., use atrous convolutions to increase the field of view without increasing the number of parameters; some of these architectures adopt the conditional random field (CRF) [32,33] as a post-processing method to optimize the pixel-wise results. In addition, an atrous spatial pyramid pooling (ASPP) module that uses multiple parallel atrous convolutions with different dilation rates is designed to learn multi-scale

contextual information in the DeepLab networks. In the architecture of PSPNet [34], the pyramid pooling module is applied to capture multi-scale features using large kernel pooling with different kernel sizes. Additionally, there are certain FCNs, such as RefineNet proposed by Lin et al. [35], which focus on methods that can fuse features at multiple scales. The above-mentioned FCNs have achieved significant improvement on open data sets or benchmarks, such as PASCAL VOC 2012 [36], Cityscape [37], and ADE20K [38]. These FCNs have also been successfully applied in remote sensing imagery segmentation for land-use identification and object detection [39] and are regarded as SOTA methods for semantic segmentation.

Compared with the semantic segmentation of natural images, there are several problems in the extraction of buildings from remote sensing images, such as the occlusion effect of building shadows, the characterization of regularized contour features of buildings, and the extremely variant and complex morphology of buildings in different areas. Thereby, the interference introduced by the foregoing factors poses a considerably difficult problem in the recovery of spatial details for the FCNs; such details are mainly manifested in the accuracy of delineation of building boundaries. To date, significant progress has been made in the extraction of building footprints from remote sensing images by means of FCNs.

To improve extraction results, many methods have been proposed to improve the SOTA models. For instance, Xu et al. [40] proposed the Res-U-Net model, in which guided filters are used to extract convolution features for building extraction. In order to obtain accurate building boundaries, the recovery of detailed information on the convolutional feature is most important. Sun et al. [41] proposed a building extraction method based on the SegNet model, in which the active contour model is used to refine the boundary. Yuan [42] designed a deep FCN that fuses outputs from multiple layers for dense prediction. In this work, a signed distance function is designed as the output representation to delineate the boundary information of building footprints. Bischke et al. [43] proposed a multi-task architecture without any post-processing stage to preserve the semantic segmentation boundary information. The proposed model attempted to optimize a comprehensive pixel-wise classification loss, which is weighted by the loss of boundary categories and segmentation labels; however, model training requires considerable time. In order to obtain the fine-grained labels of buildings, the context information of ground objects can be regarded as effective spatial constraints to predict the building labels. Liu et al. [44] proposed a cascaded FCN model, ScasNet, in which residual corrections are introduced when aggregating multi-scale contexts and refining objects. Rahklin et al. [45] introduced the Lovasz loss to train the U-Net for land-cover classification. Although the Lovasz loss has the function in improving the optimization of Intersection-over-Union (IoU) metric, it also increases the computational complexity. Another branch that can improve the extraction result is the post-processing-based methods. For example, Shrestha et al. [46] proposed a building extraction method by using conditional random fields (CRFs) and exponential linear unit. Alshehhi et al. [47] used a patch-based CNN architecture and proposed a new post-processing method by integrating the low-level features of adjacent regions. However, post-processing methods could only improve the result in a certain range, which depends on the accuracy of original segmentation results.

Based on the review of existing semantic segmentation methods above, it remains difficult to handle the balance between discrimination and detail-preservation abilities, although current methods have achieved significant improvements in accurately predicting the label of buildings or recovering building boundary information. If larger receptive fields are utilized, then more context information could be considered; however, the foregoing can make it more difficult to recover detailed information pertaining to the boundary. On the contrary, smaller receptive fields could preserve the boundary information details; however, these will lead to a substantial amount of misclassification. Meanwhile, most of the existing methods are computationally expensive, which makes their real-time application difficult.

To solve the aforementioned problem, we proposed a multi-scale context aggregation method to balance the discrimination and detail preservation abilities. On the one hand, a larger convolution

kernel is used to capture the context information; as a result, the discrimination ability of the model is enhanced. On the other hand, a larger convolution kernel is used to preserve the rich details of buildings. To fuse the different convolution features, we propose a spatial residual inception module for multi-scale context aggregation. Because of the larger convolution kernel used, the computational cost should not be neglected. In this work, we introduce depthwise separable convolutions to reduce the computational cost of the entire model. The depthwise separable convolutions can decompose a standard convolution into a combination of spatial convolutions on a single channel and a 1×1 pointwise convolution; thus, it can significantly reduce the computational complexity. In this regard, we propose a novel end-to-end FCN architecture to extract buildings from the remote sensing image. The contributions of this paper are summarized as follows.

(1) A spatial residual inception (SRI) module, which is used to capture and aggregate multi-scale contextual information, is proposed. With this module, we could improve the discriminative ability of the model and obtain more accurate building boundaries. Moreover, we introduce depthwise separable convolutions to reduce the number of parameters for high training efficiency. As a result, the computational cost could be reduced.

(2) The experimental part exhibits excellent performance on two public building labeling datasets, that is, the Inria Aerial Image Labeling Dataset [48] and Wuhan University (WHU) Aerial Building Dataset [49]. Compared to several SOTA FCN models, such as SegNet, U-Net, RefineNet, and DeepLab v3+, higher F1 scores (F1) and IoUs are achieved by our proposed SRI-Net while retaining the morphological details of buildings.

The remainder of this paper is organized as follows. In Section 2, the structure of the proposed SRI-Net is discussed in detail. Section 3 presents the comparison experiments of the proposed model and some SOTA FCNs on two open datasets. In Section 4, we further discuss certain strategies to improve the accuracy of segmentation. The conclusions are elaborated in Section 5.

## 2. Proposed Method

In this section, the proposed architecture is presented in detail. As stated above, the obstacle in semantic segmentation using FCNs is mainly the loss of detailed location information caused by pooling operations and convolutions with stride. In the application of building footprint extraction, the main problem to be solved is how to make the model adapt to the extraction of buildings with different sizes, especially large buildings. To resolve these problems, we propose a novel FCN model that can obtain rich and multi-scale contexts for finer segmentation and, at the same time, significantly reduce the number of trainable parameters and memory requirements.

### 2.1. Model Overview

As illustrated in Figure 1, our proposed model consists of two components. The first component, which generates multi-level semantics, is modified from the well-known ResNet-101. We enlarge the convolutional kernel size and adopt dilated convolutions to broaden the receptive fields; accordingly, more sufficient contexts are considered. In order to alleviate the explosive increase in parameters caused by larger convolutional kernels, we have introduced the depthwise separable convolutions for efficiency. In addition, dilated convolutions are also adopted in our encoder to capture richer contexts. Further details pertaining to the dilated convolution and depthwise separable convolution can be found in Section 2.2. The modified ResNet-101 encoder generates multi-level features, specifically, $F_2$, $F_3$, and $F_4$, which have different spatial resolutions and levels of semantics. The second component is a decoder that gradually recovers spatial information to generate the final segmentation label. In this component, the spatial residual inception (SRI) module is introduced to aggregate contexts learned from multiscale receptive fields. More details of the SRI module are presented in Section 2.3. On this basis, semantic features with low-resolutions are up-sampled using a bilinear interpolation method and, thereafter, fused with features having higher resolutions. In summary, the major innovations of

the proposed model are founded on the wider receptive fields of the encoder and the proposition of the SRI module in the decoder.

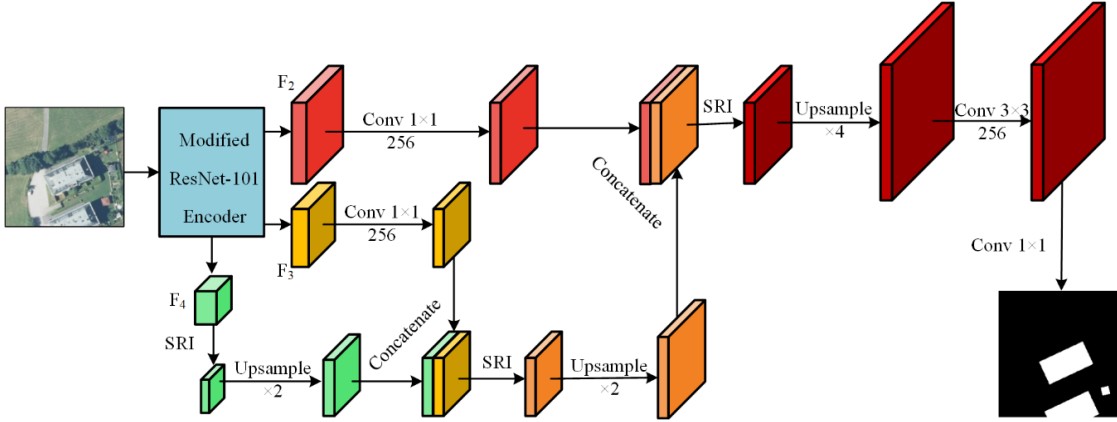

**Figure 1.** Structure of the proposed SRI-Net. The modified ResNet-101 encoder generates multi-level features, that is, $F_2$, $F_3$, and $F_4$. The spatial resolution of $F_2$, $F_3$, and $F_4$ are 1/4, 1/8, and 1/16 of the input size, respectively. The decoder employs the spatial residual inception (SRI) module to capture and aggregate multi-scale contexts and then gradually recovers the object details by fusing multi-level semantic features.

The modifications of the ResNet-101 encoder are presented in Section 2.2, where the dilated convolution and depthwise separable convolution are briefly reviewed. In Section 2.3, the SRI module is introduced in detail, and the recovery of coarse feature maps for object labels is illustrated.

### 2.2. Modified ResNet-101 as Encoder

#### 2.2.1. Dilated/Atrous Convolution

The dilated convolution, also known as atrous convolution, is a convolution with holes. As shown in Figure 2, the dilated convolution enables the enlargement of the receptive fields of filters to capture richer contexts by filling zeros between the elements of a filter. Because these filled zeros are not necessary for training, the dilated convolution can substantially expand its receptive fields without increasing computational complexity. The standard convolution can be treated as a dilated convolution with a dilation rate of 1.

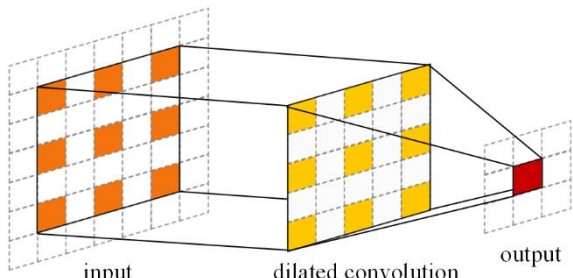

**Figure 2.** Dilated convolution with a 3 × 3 kernel and a dilation rate of 2. Only the orange-filled pixels are multiplied pixel by pixel to produce output.

#### 2.2.2. Depthwise Separable Convolution

The depthwise separable convolution is a variant of the standard convolution; it achieves a similar performance through the independent application of depthwise convolutions and pointwise convolutions independently. Specifically, the depthwise convolution refers to the spatial dimensional convolution on each channel of the input tensor, whereas the pointwise convolution applies a standard

$1\times1$ convolution to fuse the output of each channel [50], as presented in Figure 3. For example, to perform $C_o$ $k \times k$ convolutions to a tensor with $C_i$ channels, the total number of parameters using standard convolutions reaches $C_i C_o k^2$. However, in applying depthwise separable convolutions, $C_i$ $k \times k \times 1$ depthwise convolutions are performed on each channel; thereafter, $C_o$ $1 \times 1 \times C_i$ pointwise convolutions are used to aggregate outputs. As a result, the number of parameters is $C_i k^2 + C_i C_o$, which is considerably less than that of the standard convolution. It is evident that depthwise separable convolutions can significantly reduce computations. In our model, the use of depthwise separable convolutions reduces the number of parameters by more than 56 million.

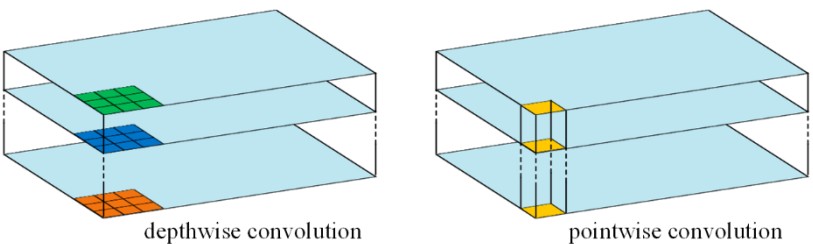

**Figure 3.** Depthwise separable convolution with $3\times3$ kernels for each input channel.

### 2.2.3. Residual Block

The most basic module of ResNet is the residual block as illustrated in Figure 4a,b. In residual blocks, the input tensor is directly or indirectly connected to the intermediate output of the residual mapping functions to avoid gradient vanishment or explosion [20,51]. Typically, there are two types of residual blocks; among these, the identity block that uses a parameter-free identity shortcut (Figure 4a) is more widely applied because of its high efficiency, and the convolutional block that employs a projection shortcut (Figure 4b) is typically used for matching dimensions [20]. In our residual blocks, input features are firstly normalized by a batch normalization (BN) [52] layer and pre-activated by a rectified linear unit (ReLU) [17] activation layer. Previous works have demonstrated that large kernels have a significant function in performing dense predictions [53]; hence, we enlarge the $3\times3$ kernels in residual blocks to $5\times5$ kernels. After the mapping of three depthwise separable convolutional layers (kernel sizes of $1\times1$, $5\times5$, and $1\times1$), feature maps are directly added to the pre-activated features in the identity block. On the other hand, in the convolutional block, the pre-activated features are transformed by a $1\times1$ convolution with stride before being added. Note that each depthwise separable convolution is followed by BN and ReLU layers. If it is assumed that the base depth of the residual block is $n$, then the detailed hyper-parameters can be obtained, as summarized in Table 1.

**Table 1.** Hyper-parameters of identity block and convolutional block for the base depth $n$.

| Residual Block Type | Depthwise Separable Convolution Index | Number of Filters | Kernel Size |
|---|---|---|---|
| | 1 | $n$ | $1\times1$ |
| Identity block | 2 | $n$ | $5\times5$ |
| | 3 | $n$ | $1\times1$ |
| | 1 | $n$ | $1\times1$ |
| Convolutional block | 2 | $n$ | $5\times5$ |
| | 3 | $4n$ | $1\times1$ |
| | shortcut | $4n$ | $1\times1$ |

### 2.2.4. Encoder Design

In the modified ResNet-101 encoder shown in Figure 4d, 64 $7\times7$ depthwise separable convolution kernels with a stride of 2 are firstly used to extract shallow features from the input image. In order to further reduce the spatial dimensions of low-level features, a max-pooling layer with a pooling size of $3\times3$ and stride of $2\times2$ is added. Thereafter, four bottleneck blocks are stacked behind the pooling layer to learn deep semantics efficiently. Each bottleneck block is composed of $m$ residual

blocks, among which the first *m-1* are identity blocks, and the last is a convolutional block (Figure 4c). The base depths (*n*) of bottleneck blocks are 64, 128, 256, and 512, and the residual blocks for the bottleneck blocks are 2, 3, 6, and 4, respectively. Note that the size of feature maps after max-pooling is approximately 1/4 that of the input image; moreover, after each bottleneck block with a stride of 2, the feature maps might be reduced by half. Therefore, the outputs of the last activation layers before the stride (activation outputs of the convolutional block in the first and second bottlenecks, that is, $F_2$ and $F_3$) are extracted as low-level features, which are approximately 1/4 and 1/8 of the input image size, respectively. The output of the encoder, that is, $F_4$, which is 1/16 of the input image size, is regarded as the high-level semantics.

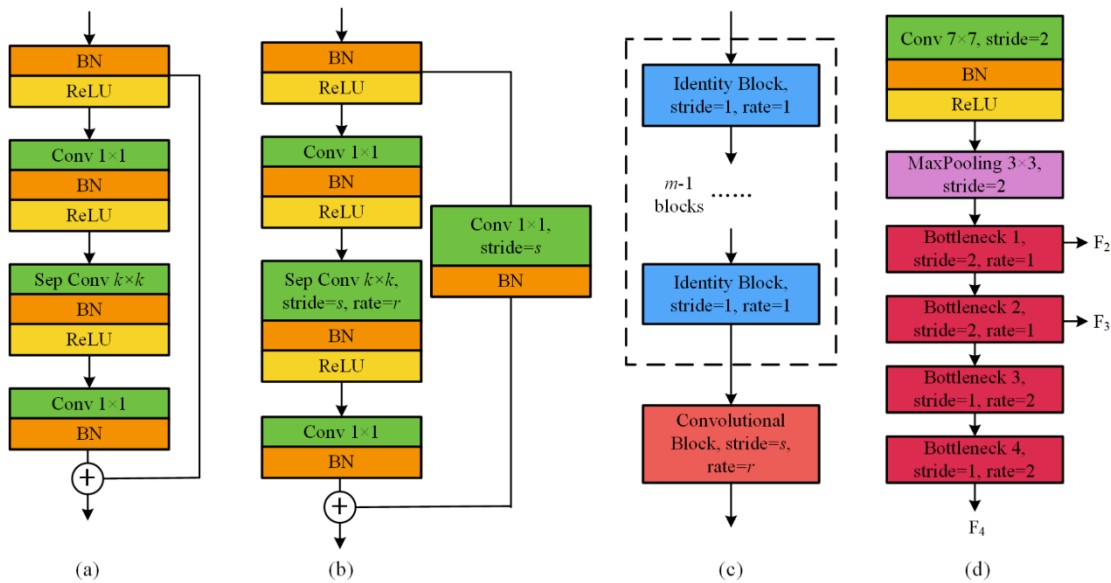

**Figure 4.** Structure of components and architecture of the modified ResNet-101 encoder. (**a**) and (**b**) are the identity block and the convolutional block, respectively, with $k \times k$ depthwise separable convolutions, stride *s,* and dilated rate *r.* (**c**) The residual bottleneck that has *m* blocks, with stride *s* and dilated rate *r.* (**d**) The modified ResNet-101 encoder. $F_2$, $F_3$, and $F_4$ represent features that are 1/4, 1/8, and 1/16 of the input image size, respectively. BN: batch normalization; ReLU: rectified linear unit.

### 2.3. Decoder with SRI Module

Before discussing the decoder structure, the newly proposed SRI module is first introduced. As illustrated in Figure 5, the SRI module is a variant of the inception block proposed in [19]. The difference is that, in the SRI module, the features learned from convolutions of different kernel sizes, that is, $3 \times 3$ and $7 \times 7$, are fused together; this significantly aids in the aggregation of multi-scale contextual information. To increase the computational efficiency, convolutions with expensive computational cost are factorized to two asymmetric convolutions. For example, a $3 \times 3$ convolution is equivalent to using a $1 \times 3$ convolution followed by a $3 \times 1$ convolution. The $7 \times 7$ convolutions are factorized in the same manner. It has been proved that this type of spatial factorization into asymmetric convolutions is computationally efficient and can ensure the same receptive field [54]. Because of the convolution factorization in the SRI module, the number of parameters of our model is reduced by approximately 5.5 million. In our SRI module, $1 \times 1$ convolutions are employed to reduce dimensions before asymmetric convolutions. Thereafter, the features of the two factorized convolution branches are concatenated to features convolved by another $1 \times 1$ convolution layer. The $1 \times 1$ convolution is added as a dimension reduction module; thereafter, the feature map is added to the activated input features. Finally, a ReLU activation is included as well.

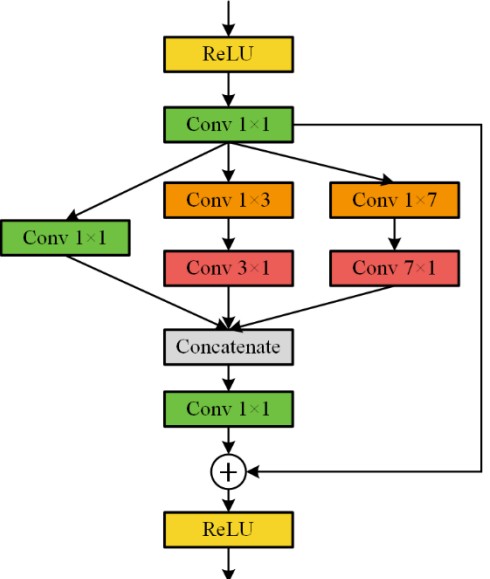

**Figure 5.** Spatial Residual Inception Module. ReLU: rectified linear unit.

The SRI module allows for learning features from multiple receptive fields; thus, the capability of capturing more contextual information is improved. As shown in Figure 4, the feature map extracted from the last bottleneck of the encoder (i.e., $F_4$ in Figure 4) is employed as the encoder output in our architecture. The $F_4$ features are first inputted into an SRI module and then up-sampled by a factor of 2. To gradually recover object segmentation details, the up-sampled features are concatenated with corresponding features that have the same spatial resolution (i.e., $F_3$ in Figure 4). We apply a 1×1 convolution with 256 channels to reduce dimensions and refine the feature map. Similarly, the recovered high-level semantics are fused with the corresponding low-level semantics (i.e., $F_2$ in Figure 4). Finally, the feature map is up-sampled and refined without any connection with low-level semantics. A sigmoid function is added before the final output, and a threshold of 0.5 is adopted to segment the probability output into binary results.

*2.4. Evaluation Metrics*

In our study, four metrics, that is, precision, recall, F1-score (F1), and IoU, are used to quantitatively evaluate the segmentation performance from multiple aspects in the following experiments. Precision, recall, and F1 are defined as:

$$\text{Precision} = \frac{TP}{TP + FP} \tag{1}$$

$$\text{Recall} = \frac{TP}{TP + FN} \tag{2}$$

$$\text{F1} = \frac{2 \cdot Precision \cdot Recall}{Precision + Recall} \tag{3}$$

where *TP*, *FP*, and *FN* represent the pixel number of true positive, false positive, and false negative, respectively. In our study, buildings pixels are positive while the background pixels are negative.

IoU is defined as:

$$\text{IoU} = \frac{\left| P_p \cap P_t \right|}{\left| P_p \cup P_t \right|} \tag{4}$$

where $P_p$ represents the set of pixels predicted as buildings, while $P_t$ is the ground truth set. $|\cdot|$ is the function to calculate the number of pixels in the set.

### 3. Experiments and Evaluations

#### 3.1. Datasets

Inria Aerial Image Labeling Dataset: This dataset is proposed by [44], which consists of orthographic aerial RGB images in 10 cities all over the world. The aerial images have tiles of $5000 \times 5000$ pixels at a spatial resolution of 0.3 m, covering about 2.25 km$^2$ per tile. The dataset covers most types of buildings in the city, including sparse courtyards, dense residential areas, and large venues. The ground truth provides two types of labels, that is, building and non-building. However, the ground truth of the dataset is only available for the training set, which consists of aerial images in 5 cities, each with 36 tiles of images. Therefore, we selected image 1 to 5 of each city from the training set for validation, and the rest were used for training.

WHU Aerial Building Dataset: This dataset is proposed by [45]. The WHU aerial dataset covers more than 187,000 buildings with different sizes and appearances in New Zealand. The dataset covers a surface area of about 450 km$^2$, and the whole aerial image and the corresponding ground truth are provided. The images have 8189 tiles of $512 \times 512$ pixels with 0.3 m resolution. The dataset was divided into three parts, among which the training set and validation set consisted of 4736 and 1036 images, respectively, while the testing set contained 2416 images.

#### 3.2. Implementation Settings

Because of the limitation in graphic processing unit (GPU) memory, the Inria aerial images and labels were simultaneously cropped to tiles of $256 \times 256$ pixels for convenience. To improve the model robustness, several transformations were employed for data augmentation, including random flipping, color enhancement, and zooming in. All pixel values were rescaled between 0 and 1. To make the model converge quickly, we initially adopted Adam optimizer [55] with a learning rate of 0.0001. The learning rate was decayed at a rate of 0.9 per epoch. To avoid over-fitting, an L2 regularization was introduced in all the convolutions with a weight decay of 0.0001. Where applicable, we accelerated the model training along with Nvidia GTX 1080 GPU.

#### 3.3. Model Comparisons

In view of the two major improvements proposed in this paper, we designed two test models by fixing the structure of the decoder and encoder for separate comparisons. One of the test models was to replace the encoder with conventional ResNet-101 on the basis of SRI-Net (referred to SRI-Net $\alpha$1). The other model was to modify SRI-Net by replacing the SRI module with the ASPP (called SRI-Net $\alpha$2). By comparing the two test models with the proposed model, we could compare and analyze the effectiveness of the two major improvements achieved in this study, that is, the modified encoder and the SRI module.

In addition to the distinct comparison experiments, non-deep learning methods (such as MBI) were introduced to the comparison for verifying the potential and application prospect of deep learning semantic segmentation models in building extraction. Moreover, the proposed SRI-Net was compared with extensive SOTA methods. In this study, the widely adopted SegNet, U-Net, RefineNet, and DeepLab v3+ were employed in the model comparison.

MBI: The MBI is a popular index proposed by Huang and Zhang [56] for building extractions. Typically, the MBI is applicable to high-resolution images with near-infrared bands, and it can also be applied to aerial images with visible bands only. Because of the operability and robustness of the MBI, several building extraction frameworks based on the MBI have been proposed since its inception [57].

SegNet: Badrinarayanan et al. [27] proposed SegNet for the semantic segmentation of road scenes. Several shortcut connections are introduced, and the indices of max-pooling in the VGG-based encoder are used to perform up-sampling. Overall, SegNet affords the advantage of memory efficiency and fast speed, and it is still the fundamental FCN architecture [58].

U-Net: Ronneberger et al. [24] proposed U-Net for biomedical image segmentation. It consists of a contracting path and a symmetric expanding path to capture contexts. The contracting path, that is, the encoder, applies a max-pooling with a stride of 2 for gradual down-sampling, whereas the expanding path gradually recovers details through a shortcut connection with corresponding low-level features and thus enables precise localization. Because of its excellent performance, U-Net, as well as its variants, has been applied to many tasks [59].

RefineNet: RefineNet is a multi-path refinement network proposed by Lin et al. [35]. It aims to refine object details through multiple cascaded residual connections. In our comparison experiments, a 4-cascaded 1-scale RefineNet with ResNet-101 as the encoder is used.

DeepLab v3+: DeepLab v3+ is a new masterpiece of DeepLab FCNs proposed by Chen et al. [26]. In DeepLab v3+, an improved xception-41 [50] encoder is applied to capture contexts, and the ASPP module is also used to aggregate multi-scale contexts. To a certain extent, DeepLab v3+ can achieve a new SOTA performance in benchmarks, such as VOC 2012.

### 3.4. Experimental Results

3.4.1. Comparison on the Inria Aerial Image Labeling Dataset

We predicted the labels of testing images with a stride of 64. The sensitivity analysis of the stride is described in detail in the discussion section. Table 2 summarizes the quantitative comparison with different models on the Inria Aerial Image Labeling Dataset. Figure 6 further demonstrates the qualitative comparison of segmentation results among the presented models except for the distinct comparison test models. It can be clearly observed that deep learning methods have significant advantages in building extraction in complex backgrounds. Although the absence of a near-infrared band introduces huge obstacles to building extractions using the MBI, objectively, the comparison results show that the method based on the morphological operator cannot cope with the automatic extraction of buildings in complex scenes. These results indicate that deep learning has a broad application prospect in managing remote sensing tasks, such as building extraction, and will definitely promote the paradigm shift in remote sensing.

The comparison results with SRI-Net $\alpha1$ and SRI-Net $\alpha2$ aid in verifying the efficiency of the proposed improvements in our model. It should be noted that both the modified ResNet-101 encoder and the decoder with the SRI module introduce slight improvements in the segmentation accuracy. These results suggest that it is a reasonable attempt to improve the performance of FCN in the direction of expanding receptive fields and considering multi-scale contexts. The summary in Table 2 also depicts the excellent performance of SRI-Net over other SOTA FCNs, such as SegNet, U-Net, RefineNet, and DeepLab v3+. It is evident that the highest F1 and IoU, that is, 83.56% and 71.76%, respectively, are obtained by SRI-Net. The performance improvement is mainly derived from the use of large convolution kernels in the encoder and the introduction of SRI module, which captures multi-scale contexts for accurate semantic mining. Among several SOTA FCNs, SegNet has the lowest accuracy and generates segmentation predictions with a more blurry boundary (row 4 in Figure 6), which significantly influences the accurate description of building footprints. The boundary of segmentation prediction results generated by U-Net and RefineNet are relatively clear, and good segmentation accuracy can be achieved. However, they remain incapable of accurately identifying large buildings (the first two columns of rows 5 and 6 in Figure 6). DeepLab v3+ and SRI-Net can achieve remarkable segmentation accuracy, provide accurate discrimination for large buildings, and identify building boundaries. However, in general, ubiquitous improvements in model architecture achieve relatively better results. The segmentation results of SRI-Net not only retain the detailed information of edges and corners of buildings but also describes the global information of building footprints.

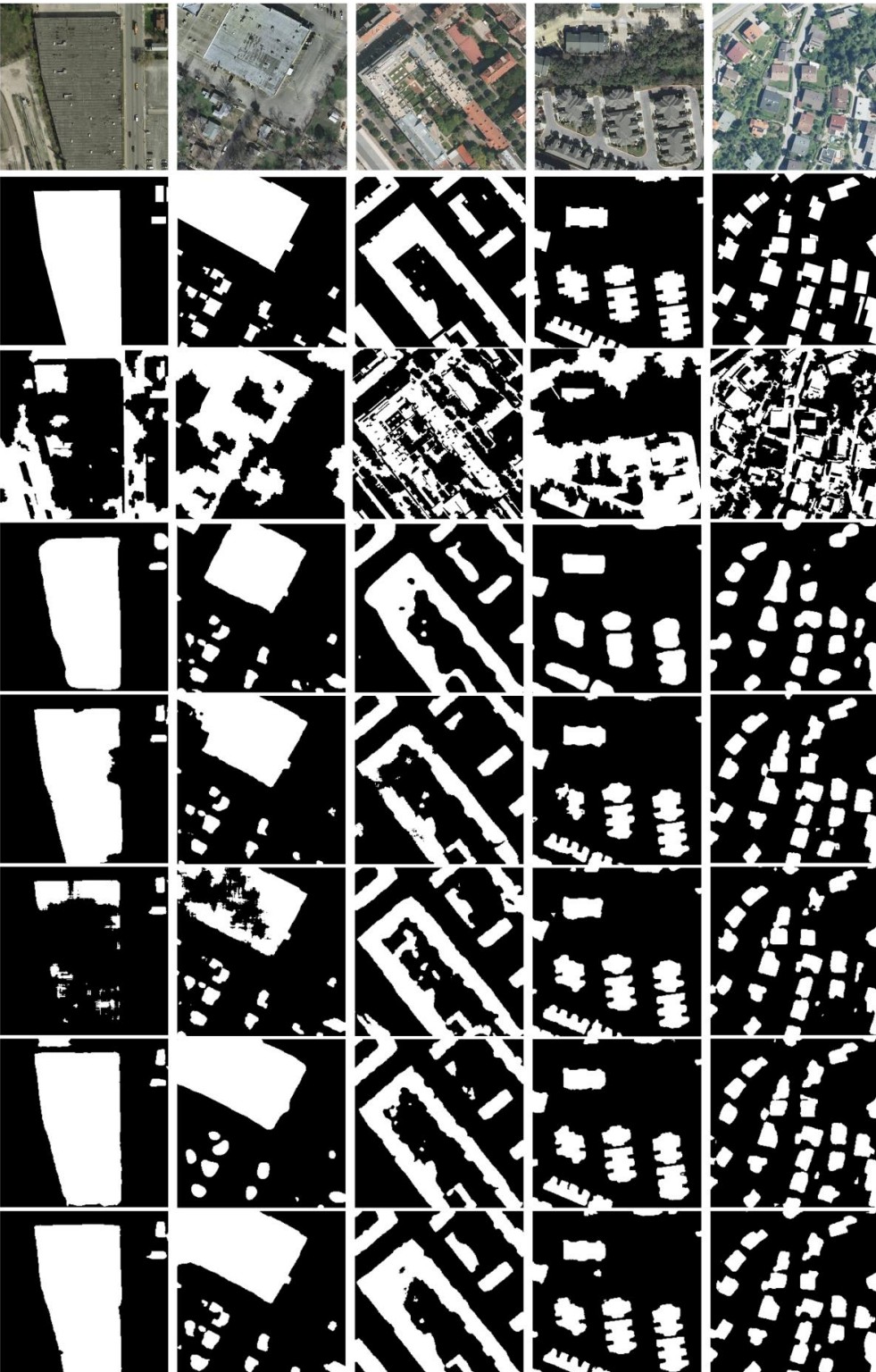

**Figure 6.** Examples of building extraction results produced by our proposed model and various state-of-the-art (SOTA) models on the Inria Aerial Labeling Dataset. The first two rows are aerial images and ground truth, respectively. Results in row 3 are produced using the morphological building index (MBI), and those of rows 4 to 8 are segmentation results of SegNet, U-Net, RefineNet, DeepLab v3+, and SRI-Net, respectively. SRI: spatial residual inception.

**Table 2.** Quantitative comparison (%) with the state-of-the-art (SOTA) models on Inria Aerial Image Labeling Dataset (best values are underlined).

|  | Precision | Recall | F1 | IoU |
|---|---|---|---|---|
| MBI | 54.68 | 50.72 | 52.63 | 35.71 |
| SegNet | 79.63 | 75.35 | 77.43 | 63.17 |
| U-Net | 83.14 | 81.13 | 82.12 | 69.67 |
| RefineNet | 85.36 | 80.26 | 82.73 | 70.55 |
| Deeplab v3+ | 84.93 | 81.32 | 83.09 | 71.07 |
| SRI-Net $\alpha$1 | 84.18 | 81.39 | 82.76 | 70.59 |
| SRI-Net $\alpha$2 | 84.44 | <u>81.86</u> | 83.13 | 71.13 |
| SRI-Net | <u>85.77</u> | 81.46 | <u>83.56</u> | <u>71.76</u> |

IoU: Intersection-over-Union; MBI: morphological building index; SRI: spatial residual inception.

### 3.4.2. Comparison on the WHU Aerial Building Dataset

As summarized in Table 3, the quantitative comparison results on the WHU Aerial Building Dataset are similar to those on the Inria Aerial Labeling Dataset. The deep learning models exhibit a considerable advantage over traditional non-deep learning methods, such as the MBI, in building footprint extraction at a fine scale. Evaluated on all the four metrics on the WHU Aerial Building Dataset, our proposed SRI-Net outperforms SegNet, U-Net, RefineNet, and DeepLab v3+ and achieves a considerably high IoU (89.09%). Compared to the results on the Inria Aerial Image Labeling Dataset, the IoU metrics are all higher than 85%, indicating that the WHU Aerial Building Dataset is of higher quality and is easier to distinguish. In fact, as discussed in [49], there are more wrong labels, high buildings, and shadows in the Inria Aerial Image Labeling Dataset that may substantially influence the discriminative ability of the FCN model. Figure 7 shows the visual performances of the comparison results. As demonstrated in the Figure, the FCN models are capable of detecting small buildings (last column in Figure 7). However, for a large building, such as those in the first two columns in Figure 7, SegNet, U-Net, and RefineNet are impotent; consequently, a certain part or an entire building is missing. Compared to DeepLab v3+, the application of successive recovery strategies leads to finer building contours (the last two rows in Figure 7) and better performance.

**Table 3.** Quantitative comparison (%) with the state-of-the-art (SOTA) models on Wuhan University (WHU) Aerial Building Dataset (best values are underlined).

|  | Precision | Recall | F1 | IoU |
|---|---|---|---|---|
| MBI | 58.42 | 54.60 | 56.45 | 39.32 |
| SegNet | 92.11 | 89.93 | 91.01 | 85.56 |
| U-Net | 94.59 | 90.67 | 92.59 | 86.20 |
| RefineNet | 93.74 | 92.29 | 93.01 | 86.93 |
| Deeplab v3+ | 94.27 | 92.20 | 93.22 | 87.31 |
| SRI-Net $\alpha$1 | 93.37 | 92.43 | 92.90 | 86.74 |
| SRI-Net $\alpha$2 | 94.47 | 92.26 | 93.35 | 87.53 |
| SRI-Net | <u>95.21</u> | <u>93.28</u> | <u>94.23</u> | <u>89.09</u> |

IoU: Intersection-over-Union; MBI: morphological building index; SRI: spatial residual inception.

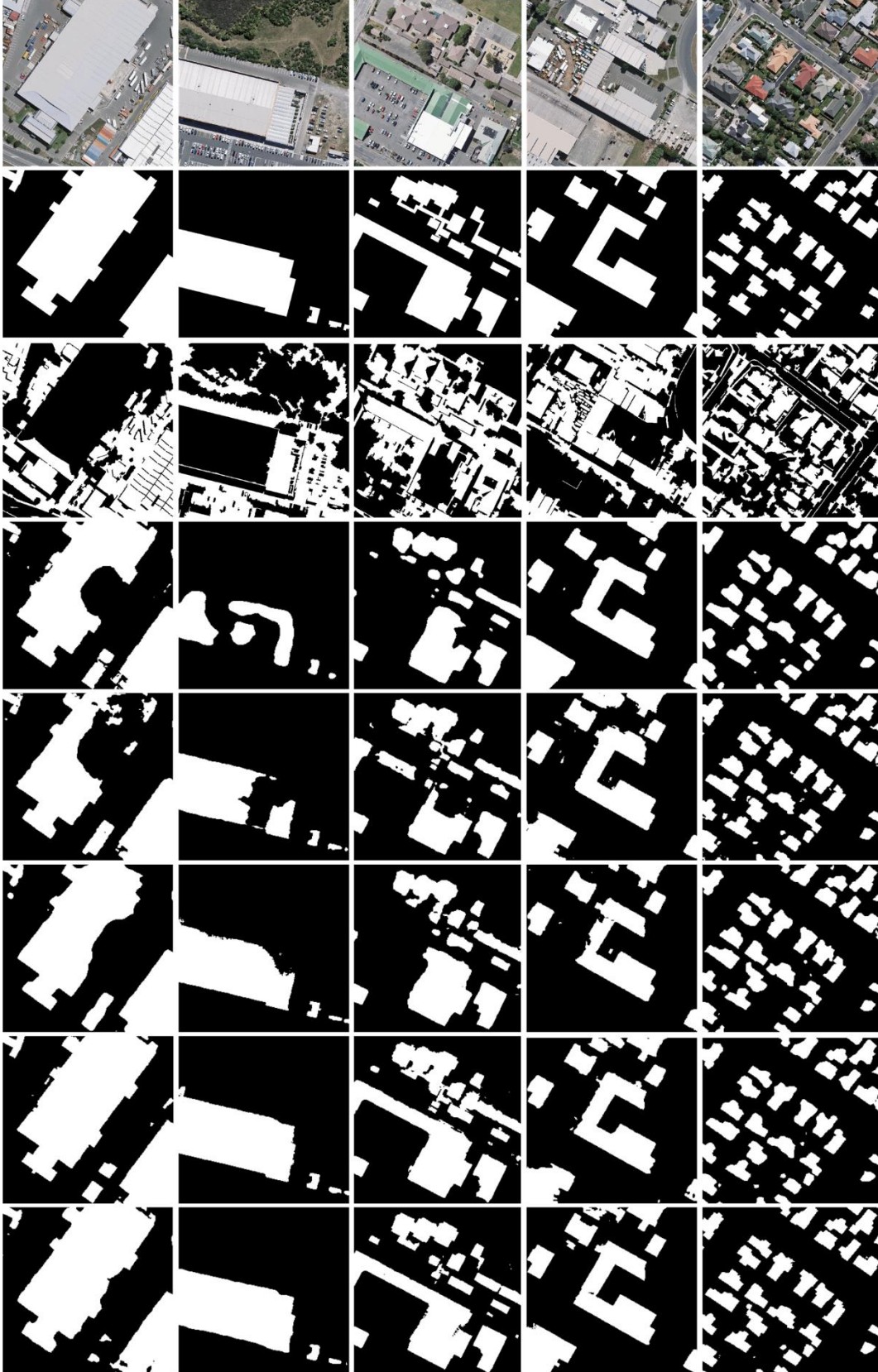

**Figure 7.** Examples of building extraction results produced by our proposed model and various state-of-the-art (SOTA) models on the Wuhan University (WHU) Aerial Building Dataset. The first two rows are aerial images and ground truth, respectively. Results in row 3 are produced using the morphological building index (MBI), and that of rows 4 to 8 are segmentation results of SegNet, U-Net, RefineNet, DeepLab v3+, and SRI-Net, respectively. SRI: spatial residual inception.

In summary, SegNet uses the VGG-based encoder and utilizes the indices of max-pooling to perform up-sampling. This type of symmetrical and elegant architecture fails to effectively capture multi-level semantic features and recover object details, While U-Net and RefineNet can attain better performances by capturing high-level features and efficiently recovering details by integrating multi-level semantics. However, they still fail to consider multi-scale contexts; consequently, it is considerably difficult to distinguish objects with different sizes, especially large buildings. The ASPP and SRI module have successfully realized feature-capturing in receptive fields with different sizes; thus indicating that they have strong flexibility in identifying objects of different sizes. In SRI-Net, the features are successively recovered through three times of $2\times$ up-sampling. Thus, compared with DeepLab v3+, a slight improvement in accuracy is achieved. Moreover, the introduction of large convolution kernels can also provide a larger receptive field for capturing richer contextual information. SRI-Net can mine and fuse multi-scale high-level semantic information accurately and has high flexibility for detecting buildings of different sizes, which can be attributable to the SRI module. Moreover, SRI-Net can accurately depict the boundary information of buildings and distinguish objects that are easily misidentified (for example, containers).

## 4. Discussion

### 4.1. Influence of Stride on Large Image Prediction

As stated in the previous sections, image cropping will inevitably result in distinct marginal stitching seams. Because of the lack of sufficient contextual information in the marginal area when convolutions are applied, zero padding is typically used and thus brings evident inconsistencies with the original image. Additionally, objects, such as buildings, might be split by cropping the images. Therefore, predicting with a stride is typically an advisable approach that considers the context of pixels in the marginal area [49]. Undoubtedly, a small stride would improve the accuracy of segmentation, but it would necessitate considerable computational time. Consequently, it is crucial to determine the proper stride for the fast and accurate detection of building footprints, particularly for automatic building extraction in large areas.

For comparison, we evaluated the IoU and computation time on the testing set of Inria Aerial Image Labeling Dataset at different stride values. Figure 8 shows that the smaller the stride is adopted, the more accurate and refined the results are obtained. This is not unexpected because the pixels are predicted more frequently in different scenes as smaller strides are used, which can avoid the loss of contextual information caused by scene segmentation. Figure 9 further presents the probability maps using different stride values. The whiter pixels indicate a higher probability of images being buildings. It can be observed that large stride values may result in many patches that can be mistaken as buildings (first row) and parts of large buildings being omitted (second row). However, the stride value only has a slight influence on the prediction of compact and small buildings (third row). A relatively high and stable IoU can be achieved when the stride is 64. When the stride value is further halved to 32, the computational time is increased to four times the previous consumption, but there is only a slight improvement in the IoU. For rapid and accurate building detection, we recommend the use of 64, that is, 1/4 of the input image size.

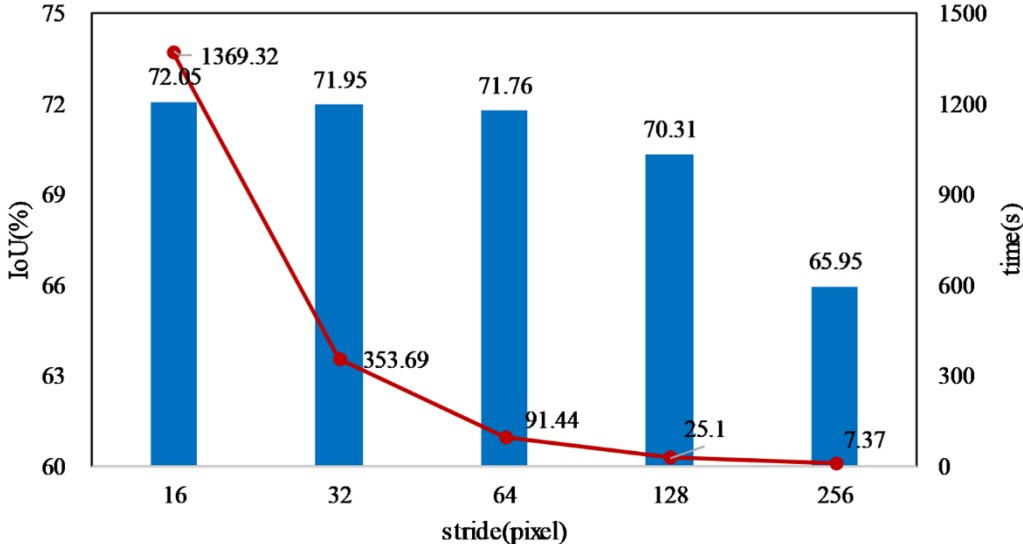

**Figure 8.** Intersection-over-Union (IoU) and computational time consumed using different stride values in the prediction of large images.

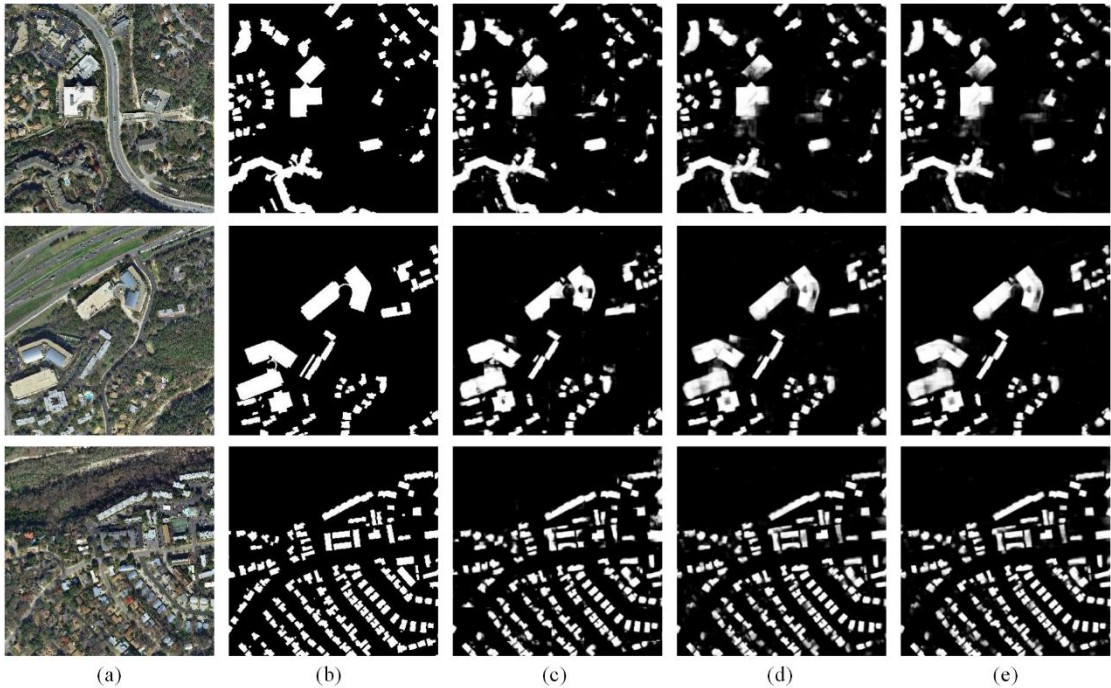

**Figure 9.** Predictions on Inria Aerial Image Labeling Dataset using different strides. (**a**) Aerial image. (**b**) ground truth, (**c**) stride=256, (**d**) stride=64, (**e**) stride=16, (**c**)–(**e**) are building probability maps.

### 4.2. Pretrained ResNet-101 on the University of California (UC) Merced Land Use Dataset

Because of the limited training data, it may be difficult to improve the accuracy of the model by directly training it on the dataset. Previous studies have shown that a pre-training strategy has a positive effect on improving the performance of the model [26,60]. Considering the characteristics of aerial images, we concatenated a global average pooling layer behind our proposed encoder (i.e., the modified ResNet-101) and trained a classification model in the UC Merced Land Use Dataset [61]. Thereafter, we initialized the encoder weights using the pre-trained model and retrained our SRI-Net on the two above-mentioned building labeling datasets. Table 4 summarizes the performances of the model that have been improved by initializing the weights of the encoder using the pre-trained

weights. Specifically, the IoU improvement over the Inria dataset is 1.59%, whereas the performance improvement over the WHU dataset is only 0.14%. The possible reason is that the WHU dataset is already extremely accurate, but the training data is not enough. Although the improvements are not sufficiently significant, it also indicates that using the pre-trained weights is a reasonable solution to boost segmentation performance.

**Table 4.** Quantitative comparison of evaluation metrics between directly training and pre-training on University of California (UC) Merced Land Use Dataset.

| Dataset | Strategy | Precision | Recall | F1 | IoU |
|---|---|---|---|---|---|
| Inria Aerial Image Labeling Dataset | Direct training | 95.97 | 96.02 | 95.97 | 71.76 |
| | Pre-training | 96.44 | 96.53 | 96.44 | 73.35 |
| WHU Aerial Building Dataset | Direct training | 95.21 | 93.28 | 94.23 | 89.09 |
| | Pre-training | 95.67 | 93.69 | 94.51 | 89.23 |

IoU: Intersection-over-Union; WHU: Wuhan University.

## 5. Conclusions

In this paper, a new FCN model (SRI-Net) is proposed to perform semantic segmentation on high-resolution aerial images. The proposed method mainly focuses on two key innovations. (1) Larger convolutional kernels and dilated convolutions are used in the backbone of the SRI-Net architecture to obtain a wider receptive field for context mining, which enables learning rich semantics for detecting objects in complex backgrounds. Moreover, depthwise separable convolutions are introduced to improve computational efficiency without reducing performance. (2) A spatial residual inception (SRI) module is proposed to capture and aggregate contexts from multi-scales. Similar to the ASPP, the SRI module captures features in parallel by considering contexts from different scales, thus significantly improving the performance of SRI-Net. With reference to the concept of Inception module, large kernels in the SRI module are factorized, thereby reducing the parameters of the module. Features with coarse resolutions but high-level semantics are successively integrated and refined with low-level features. Hence, object details, such as boundary information, can be recovered and depicted well.

Several experiments conducted over two public datasets have further highlighted the advantages of SRI-Net. The comparison results demonstrate that SRI-Net can obtain segmentation that is remarkably more accurate than the SOTA encoder-decoder structures, such as SegNet, U-Net, RefineNet, and DeepLab v3+, while retaining detailed and global information of buildings. Furthermore, large buildings, which may be omitted or misclassified by SegNet, U-Net, or RefineNet, can also be precisely detected by SRI-Net because of the richer and multi-scale contextual information. The outstanding performance of SRI-Net provides a signal that can be applied to the extraction and change detection of buildings in large areas. Moreover, land covers, such as tidal flat, water area, snow cover, and even specific forest species, can be accurately extracted for dynamic monitoring by means of semantic segmentation. However, remote sensing images, including high-resolution aerial images, are extremely complex in context, and the existence of shadows, occlusions, and high buildings also introduces considerable problems to the semantic segmentation and extraction of buildings. In addition, how to make segmented buildings maintain their unique morphological characteristics, such as straight lines and right angles, is also a problem that requires immediate resolution.

**Author Contributions:** All authors made significant contributions to this article. Conceptualization, P.L. and Q.S.; Methodology, P.L., Q.S., and M.L.; Formal analysis, P.L. and Y.Z.; Resources, Q.S. and X.L.; Data curation, P.L., Q.S., and M.L.; Writing—original draft preparation, P.L.; Writing—review and editing, P.L., Q.S., M.L., J.Y., X.X., and Y.Z.; Funding acquisition, Q.S.

**Funding:** This research was supported by the National Key R&D Program of China (Grant No. 2017YFA0604402), the Key National Natural Science Foundation of China (Grant No. 41531176), and the National Nature Science Foundation of China (Grant No.61601522).

**Conflicts of Interest:** The authors declare no conflict of interest.

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
