# Peer review of "Building Footprint Extraction from High-Resolution Images via Spatial Residual Inception Convolutional Neural Network"

_remotesensing, doi:10.3390/rs11070830_

Round 1

Reviewer 1 Report

Authors compared the building detection performance of the improved deep learning network with other deep learning models. It will be useful to compare the performance with traditional methods (i.e. non deep learning based methods, such as Gilani et al  2016, ‘An automatic building extraction and regularisation technique using lidar point cloud data and orthoimage’, Remote Sensing, vol. 8, no. 3, pp. 258).

Author Response

Authors compared the building detection performance of the improved deep learning network with other deep learning models. It will be useful to compare the performance with traditional methods (i.e. non deep learning based methods, such as Gilani et al  2016, ‘An automatic building extraction and regularisation technique using lidar point cloud data and orthoimage’, Remote Sensing, vol. 8, no. 3, pp. 258).

Response: Many thanks for your suggestions. And we think your suggestion is very helpful to improve the quality of this paper. Because we do not have the Lidar point cloud data, so we could not compare the methods adding the Lidar point data. The morphological building index(MBI) is a state of art traditional method in building extraction. Thus we added MBI in our comparing experiments. The quantitative comparison can be found in Table 2 and Table 3.

Reviewer 2 Report

In this paper, the authors proposed a new FCN architecture (e.g., SRI-Net) for building extraction from high-resolution images. There are mainly two innovations of the proposed SRI-Net: (1) Larger convolutional kernels and depthwise separable convolutions was introduced to build a novel ResNet-101 encoder. (2) Spatial Residual Inception module was applied to capture and aggregate multi scale contexts for better performance. The proposed model was tested on the Inria and WHU building labeling dataset and outperformed the SegNet and U-Net.

Overall, the authors proposed a novel FCN architecture for semantic segmentation. This paper is well organized and easy to follow, and the figures and result analysis are very clear. However, there are still some questions confusing me, which are listed below. Therefore, I recommend that this paper should be considered for publication in RS after a major revision.

Major Points:

The authors have compared the proposed SRI-Net with two SOTA FCNs (SegNet, U-Net). However, as far as I know, both SegNet and U-Net were proposed in the early development of FCN. In the recent 3 years, FCNs like PSPNet and RefineNet have been proposed and achieved new SOTA performances in competitions including the VOC2012 segmentation task. Thus, I strongly recommend the authors to add comparison experiments with PSPNet and RefineNet.

Minor points:

1. Line 125~139. When introducing the contribution of this paper, the problems/shortcomings of existing studies should be concluded. The authors reviewed the proceedings using FCNs for building extraction but didn’t summarize the core problems corresponding to the contribution.

2. There are several incorrect cross references to figures, such as line 161, 166, 182 and 259.

3. In table 4, IoU improvement of the pre-trained model on the Inria is 1.59%, but not 1.02%. I request the authors to check the data carefully.

4. In section 4.2, the encoder of the SRI-Net was trained on UC Merced Land Use Dataset for classification task, which could be considered as a transfer learning strategy. However, transfer learning is usually employed to transfer knowledge between similar tasks. Here, the encoder of a semantic model was pre-trained in a classification task. Please explain why it works.

Author Response

Point 1: The authors have compared the proposed SRI-Net with two SOTA FCNs (SegNet, U-Net). However, as far as I know, both SegNet and U-Net were proposed in the early development of FCN. In the recent 3 years, FCNs like RefineNet have been proposed and achieved new SOTA performances in competitions including the VOC2012 segmentation task. Thus, I strongly recommend the authors to add comparison experiments with RefineNet.

R: Many thanks for your suggestions. In the experimental part, we add RefineNet and Deeplab v3+, which are state of art methods in semantic segmentation field. The compared result can be shown in Figure 6,7 and Table 2,3.

Point 2: Line 125~139. When introducing the contribution of this paper, the problems and shortcomings of existing studies should be concluded. The authors reviewed the proceedings using FCNs for building extraction but didn’t summarize the core problems corresponding to the contribution.

R: Many thanks for your suggestion. The conclusion can be draw as follow:

Based on the review of existing semantic segmentation methods above, it remains difficult to handle the balance between discrimination and detail-preservation abilities although current methods have achieved significant improvements in accurately predicting the label of buildings or recovering building boundary information. If larger receptive fields are utilized, then more context information could be considered; however, the foregoing can make it more difficult to recover detailed information pertaining to the boundary. On the contrary, smaller receptive fields could preserve the boundary information details; however, these will lead to a substantial amount of misclassification.

Point 3 There are several incorrect cross references to figures, such as line 161, 166, 182 and 259.

R: Many thanks for your careful check. We have revised this wrong cross references.

Point 4: In table 4, IoU improvement of the pre-trained model on the Inria is 1.59%, but not 1.02%. I request the authors to check the data carefully.

R: Many thanks for your careful check. In new manuscript, we have rewritten experimental part.

Point 5: In section 4.2, the encoder of the SRI-Net was trained on UC Merced Land Use Dataset for classification task, which could be considered as a transfer learning strategy. However, transfer learning is usually employed to transfer knowledge between similar tasks. Here, the encoder of a semantic model was pre-trained in a classification task. Please explain why it works.

R: Many thanks for your question. The encoder part of the semantic segmentation play the role of extracting features which have proved can be transferred to similar tasks in scene classification. And the decoder part are used to recover the detail information in the convolution feature. Thus the encoder part play an important role to transfer the features to the other domain.

Reviewer 3 Report

Authors adopted a deep learning method to extract building footprints from high-resolution aerial images.

Although the idea behind this paper sounds interesting, the methodology is poorly presented and it is not clear what are the main goals and innovations behind this work. There are many grammatical errors in the text. No structure. There are several sentences written in Chinese in line 161, 166, and 182. 

The novelty of the work is limited as there are several articles do the same approach in the literature. 

There is no comparison with SOTA article liked

Wen, Qi, et al. "Automatic Building Extraction from Google Earth Images under Complex Backgrounds Based on Deep Instance Segmentation Network." Sensors 19.2 (2019): 333.

Ding, Z., et al. "STUDY ON BUILDING EXTRACTION FROM HIGH-RESOLUTION IMAGES USING MBI." International Archives of the Photogrammetry, Remote Sensing & Spatial Information Sciences 42.3 (2018).

...

Author Response

Very sorry about these mistakes. We have rewritten this paper after self-examination. The grammatical errors have been corrected. And the structure of this paper has been reorganized. 

Round 2

Reviewer 2 Report

I have no more comments.

Reviewer 3 Report

Authors modified the MS based on comments of reviewers.